# Entomopathogenic Nematodes for the Management of Plum Curculio in Highbush Blueberry

**DOI:** 10.3390/biology11010045

**Published:** 2021-12-29

**Authors:** Ana Luiza Sousa, Cesar Rodriguez-Saona, Robert Holdcraft, Vera Kyryczenko-Roth, Albrecht M. Koppenhöfer

**Affiliations:** 1Department of Entomology, Rutgers University, New Brunswick, NJ 08901, USA; avs102@sebs.rutgers.edu; 2P.E. Marucci Blueberry and Cranberry Center, Department of Entomology, Rutgers University, Chatsworth, NJ 08019, USA; crodriguez@njaes.rutgers.edu (C.R.-S.); rob.holdcraft@rutgers.edu (R.H.); vera.kyryczenko@rutgers.edu (V.K.-R.)

**Keywords:** *Conotrachelus nenuphar*, biological control, blueberry, *Vaccinium corymbosum*, *Steinernema riobrave*

## Abstract

**Simple Summary:**

The plum curculio, a snout beetle native to North America, is one of the most important fruit-feeding pests of cultivated blueberry in New Jersey. Application of certain entomopathogenic nematode (EPN) species has shown efficacy in plum, peach, and apple orchards when targeting the larval stage of plum curculio in soil. Prior to our research, however, EPNs have never been tested for control of this pest in highbush blueberries. In 2020, laboratory and field studies were conducted to: (1) determine the persistence of different EPN species in acidic blueberry soil; (2) compare the virulence of these EPNs to plum curculio larvae and pupae; and (3) compare the efficacy of these EPNs to control this pest in blueberry fields. In 2021, field studies were conducted to confirm the efficacy of one of the EPN species. *Steinernema riobrave* persisted better in blueberry sand, was the most virulent EPN against plum curculio larvae and pupae and was highly efficacious for suppression of larvae and pupae of this pest in blueberry fields. *Steinernema riobrave* has the potential to become an important component in the management of plum curculio in highbush blueberry.

**Abstract:**

*Conotrachelus nenuphar* Herbst (Coleoptera: Curculionidae) is a key pest of stone and pome fruits in the United States. Application of certain entomopathogenic nematode (EPN) species has shown efficacy in some crops when targeting the larval stage of *C. nenuphar* in soil. To date, however, no EPNs have been tested for the control of this pest in highbush blueberries. In 2020, laboratory and field studies were conducted to: (1) determine the persistence of *Steinernema riobrave*, *S. carpocapsae*, *S. feltiae*, and *Heterorhabditis bacteriophora* in acidic blueberry soil; (2) compare the virulence of these EPNs to *C. nenuphar* larvae and pupae; and (3) compare the efficacy of these EPN species to control this pest in blueberry fields. The greatest persistence in blueberry soil was exhibited by *S. riobrave* followed by *S. carpocapsae*. Superior virulence was observed in *S. riobrave* against *C. nenuphar* larvae and pupae. Promising levels of virulence were also observed in *S. carpocapsae* and *S. feltiae* against the larvae, but *S. scarabaei* had low virulence. In the field, *S. riobrave* provided significantly higher levels of *C. nenuphar* suppression (90%) than the other EPNs. The field efficacy of *S. riobrave* against *C. nenuphar* at low and high rates was confirmed in 2021. *Steinernema riobrave* has the potential to become an important component in the management of *C. nenuphar* in highbush blueberry.

## 1. Introduction

The plum curculio, *Conotrachelus nenuphar* Herbst (Coleoptera: Curculionidae), is a major pest of several fruit crops (peach, apple, plum, cherry, and blueberry) in North America [1,2,3]. In early spring, overwintered adult weevils emerge and usually begin moving from wooded areas into cultivated crops to feed, mate, and oviposit in fruits. Damage to fruit results from feeding and oviposition scars produced by *C. nenuphar* adult females and from larval burrowing. Internal larval feeding can cause early fruit drop, which typically occurs at or near the molt to the fourth-instar larvae. Infested blueberries often turn prematurely blue before experiencing early drop [4]. Fruit drop increases survivorship of the larvae that exit the fallen fruit when fully mature (fourth instar) and burrow into the soil (2.5–5 cm) to pupate [1,5]. Complete time in the soil can be as long as 30 days, with approximately eight of those days spent as a teneral adult [6,7]. After emergence, adults feed on fruits and migrate to overwintering sites in forested areas and unmanaged orchards or surrounding areas [1,7,8].

Above-ground applications of insecticides continues to be the main management tactic used to suppress *C. nenuphar* adults [9]. Due to environmental and regulatory concerns, approaches need to be developed to control *C. nenuphar* that reduce the overall amount of synthetic insecticide used such as the application of entomopathogenic nematodes (EPNs), targeting *C. nenuphar* stages in the soil [10,11,12,13,14].

Entomopathogenic nematodes, *Steinernema* spp. and *Heterorhabditis* spp., are obligate parasites of insects. The free-living EPN stage, the infective juvenile (IJ), enters hosts through natural openings (mouth, anus, spiracles, and cuticle), and after entering the host’s hemocoel, releases symbiotic bacteria, which are primarily responsible for killing the host within 24–48 h. The nematodes complete 1 to 3 generations within the host, whereafter IJs exit the cadaver to search out new hosts [15].

EPNs are effective at controlling a variety of economically important pests, including the larvae of several weevil species (Coleoptera: Curculionidae) that spend a large portion of their life cycle in the soil [16]. Previous studies have already indicated the potential of several EPN species in the genus *Steinernema* to suppress soil-dwelling stages of *C. nenuphar* in apple, plum and peach orchards, with *Steinernema riobrave* Cabanillas, Poinar, and Raulston being the most virulent species, demonstrating a significant reduction in *C. nenuphar* emergence in all field trials [11,12,13,14,17,18,19]. EPN applications tend to be most efficacious to soil or cryptic habitats due to the nematode’s sensitivity to desiccation and ultraviolet (UV) radiation [15]. Therefore, fourth-instar larva, pupa, and adult of *C. nenuphar* are potential targets for EPNs, since these stages occur in or on the soil [1].

To date, no EPNs have been tested for the control of *C. nenuphar* in highbush blueberries (*Vaccinium corymbosum* L.). In this study, we conducted laboratory and blueberry field experiments to: (1) determine the persistence of the EPNs *S. riobrave*, *Steinernema carpocapsae* (Weiser), *Steinernema feltiae* (Filipjev), and *Heterorhabditis bacteriophora* Poinar in acidic blueberry soil; (2) compare the virulence of these EPNs to *C. nenuphar* larvae and pupae; and (3) determine the efficacy of *S. riobrave*, *S. carpocapsae*, *S. feltiae*, and *Steinernema scarabaei* Stock & Koppenhöfer -to control this pest in highbush blueberry fields.

## 2. Materials and Methods

### 2.1. Laboratory Experiments

#### 2.1.1. Soils, Insects, Nematodes

Substrates collected for studying the effect of soil type on EPN persistence were a sandy loam (61% sand, 27% silt, 12% clay; 2.3% organic matter (OM); pH 5.9) and an acidic blueberry sand (93% sand, 7% silt, 0% clay; 5.1% OM; pH 3.9). The substrates were air-dried, sieved through no. 5 (4 mm openings) and no. 10 (2 mm openings) sieves, pasteurized (3 h at 72 °C), and aerated for at least 1 week before use [20]. Soil moisture release curves were established using the filter paper method for determination of soil matric potential [21].

Commercial strains of *S. carpocapsae* (commercial product Millenium^®^,BASF, Research Traingle Park, NC, USA) and *S. feltiae* (Nemasys^®^) were obtained from BASF (Research Triangle Park, NC, USA) and of *H. bacteriophora* (Nema-green^®^) from e~nema (Schwentinental, Germany); *S. riobrave* (strain 355) was obtained from Dr. David Shapiro-Ilan (USDA-ARS, Byron, GA, USA). *Steinernema scarabaei* AMK001 originated from our EPN collection (Turfgrass Entomology Lab, New Brunswick, NJ, USA).

To standardize IJ quality for experiments, the EPNs were cultured in last instar larvae of the greater wax moth, *Galleria mellonella* (L.) that were obtained from Rainbow Mealworms (Compton, CA, USA). The emerging IJs were harvested over a period of 7–14 days from emergence traps and stored in filtered tap water in tissue culture flasks at 8 °C for 7–21 days before use. Fresh starter cultures were obtained every year for the experiments and EPNs were only reared once through wax moth larvae to avoid potential modification of the commercial strains during laboratory culture. *Conotrachelus nenuphar* adults were collected in early spring using beating sheets from unsprayed blueberry fields and peach orchards located at a commercial organic blueberry farm (Hammonton, NJ, USA) and at the Rutgers Agricultural Research and Extension Center (Bridgeton, NJ, USA), and taken to the Blueberry/Cranberry Entomology laboratory at the Philip E. Marucci Center for Blueberry and Cranberry Research and Extension of Rutgers University, Chatsworth, NJ, USA. Green thinning apples were placed in sweater boxes with *C. nenuphar* adults for oviposition and starting the *C. nenuphar* larvae rearing at room temperature (22–25 °C) and a photoperiod of 14:10 (L:D) h following methods in Amis and Snow [22].

#### 2.1.2. Nematode Persistence in Different Soil Types

Blueberry sand was prepared at −10 kPa soil water potential (10% soil moisture *w*/*w*) and sandy loam at −14 kPa (14% soil moisture *w*/*w*) and the substrates were filled into 200-mL plastic cups (55 mm height × 75 mm diam). Each cup received 100 g moist substrate. The substrate was uniformly compacted to a depth of approximately 25 mm by tapping the cups and lightly pressing on the substrate surface. Two hundred IJs of *S. carpocapsae*, *S. feltiae*, *S. riobrave*, and *H. bacteriophora* were added in 250 µL tap water to the substrate surface. Then the cups were placed into plastic boxes with wet paper towel to reduce moisture loss and stored at room temperature. Every 7 days, the cups were opened for air exchange, weighed, and water was added if the weight of the cup had decreased. EPN persistence in the soil was assessed by saturation baiting with last instar wax moth larvae [23]. At 0, 7, 14, 28, and 56 days after treatment (DAT), five cups from each substrate type were opened, the soil loosened with a scupula, 10 wax moth larvae added, and the cups sealed again. Every 3 days, dead wax moth larvae were replaced with live ones and the baiting concluded after 12 days (4 baiting rounds) after the first larvae were added. Dead larvae were incubated for an additional 6 days and then dissected to confirm EPN infection. For each cup, the total number of infected larvae was recorded [20]. The experiment was conducted twice at room temperature (22–25 °C).

#### 2.1.3. Nematodes Virulence of against *C. nenuphar* Larvae and Pupae

The experiments were conducted at room temperature (22–25 °C) and the blueberry soil was prepared at −10 kPa soil water potential. For the *C. nenuphar* larvae experiment, each 200-mL plastic cup (55 mm height × 75 mm diam) received 100 g moist blueberry soil. Five *C. nenuphar* fourth-instar larvae were added per cup and the cups sealed with translucid plastic lids. Larvae that did not enter the soil within 1 h were replaced. *Steinernema carpocapsae*, *S. feltiae*, *S. riobrave*, and *S. scarabaei* were applied in 250 µL tap water to the substrate surface at the rates of 100, 200, and 400 IJs/larva. Untreated control cups received water only. At 7 and 14 DAT, the cups were checked, and infected larvae were recovered, rinsed in water, and dissected immediately to confirm EPN infection. Each EPN species and rate had 7 replicates and the experiment was conducted twice.

For *C. nenuphar* pupae, two different experiments were conducted using 30 mL plastic cups (3.3 cm height × 4 cm diam) filled with 20 g of moist blueberry soil. One pupa was released onto the soil surface and the vials were sealed with a translucid plastic lid. Pupae that did not enter the soil within 1 h were replaced. In the first experiment, *S. carpocapsae*, *S. feltiae*, *S. scarabaei*, and *H. bacteriophora* were applied in 250 µL tap water to the substrate surface at 400 IJs/pupa. Untreated control cups received water only. Each treatment had 36 vials as replicates and the experiment was conducted twice. In the second experiment, *S. riobrave* was applied in 250 µL tap water to the substrate surface at 100, 200, and 400 IJs/pupa. Untreated control cups received water only. Each treatment had 40 vials and the experiment was conducted twice. For both experiments, the cups were checked at 7 and 14 DAT, and infected pupae were recovered, rinsed in water, and dissected immediately to confirm EPN infection.

### 2.2. Field Experiment: 2020

The field experiment was conducted in an organically maintained blueberry field at the Philip E. Marucci Center, Chatsworth, NJ, USA. Highbush blueberries (25-year-old variety ‘Bluecrop’) were spaced 0.61 m between bush crowns and 5.49 m between rows center-to-center.

Four EPN species were tested. *Steinernema riobrave*, *S. carpocapsae*, and *S. feltiae* were selected based on results from the persistence and virulence laboratory experiments. *Steinernema scarabaei* AMK001 was included because it can provide excellent control, potentially even long-term suppression of another important pest of highbush blueberries: larvae of the oriental beetle, *Anomala orientalis* (Waterhouse) [24]. Due to its potential for long persistence, *S. scarabaei* could control *A. orientalis* larvae even though they would not be present yet at the appropriate timing for *C. nenuphar* EPN applications [20,25].

Experimental arenas to determine the efficacy of the EPNs consisted of 45-cm diam circular plots (1590 cm^2^) placed between two adjacent blueberry bushes within a row in the blueberry field. On 23 June 2020 (10:00–12:00 EST, sunny, 25–30 °C), the plots were treated with approximately 79,500 IJs of a given EPN species (50 IJs/cm^2^) applied by hand sprayer in a volume of 36 mL per plot. The EPNs were then watered in with 1000 mL of water (6.3 mm water) using a watering can. Then 100 blueberries infested with *C. nenuphar* larvae in the laboratory were evenly distributed on the soil surface of each plot before the plots were covered with pyramidal emergence cages (Figure 1), as described by Piñero et al. [18]. The pyramidal emergence cages (43.7 cm diameter base; 64 cm tall) were made of aluminum window screen and the top (white) portion of a boll weevil trap (ISCA technologies, Riverside, CA, USA) was fitted with a stainless-steel hose clamp at the top of each cage. This permitted the capture of adult *C. nenuphar* that, upon emergence, walked upward onto the interior of the capturing boll weevil trap. The edges of the cages were buried about 5 cm deep in the soil to prevent *C. nenuphar* adults from escaping.

Treatments were a water-only control, *S. scarabaei*, *S. riobrave*, *S. feltiae*, and *S. carpocapsae* and there were six cages per treatment. To determine EPN efficacy, cages were inspected twice weekly for 6 weeks (10 August 2020), and any adults emerged collected. The total number of *C. nenuphar* adults emerged per cage was compared among treatments. Weather data were obtained from the weather station at Chatsworth, NJ, USA (39°42′55.97″ N, 74°30′39.24″ W) (Office of the New Jersey State Climatologist; Rutgers University, New Brunswick, NJ, USA), and downloaded from the New Jersey Weather and Climate Network website (www.njweather.org/ accessed on 10 September 2021).

Arenas for the determination of EPN persistence consisted of 30 × 30 cm area directly adjacent to the emergence cages between the blueberry bushes (Figure 1). EPNs were applied at the same time as for the efficacy experiment at the same rate (50 IJs/cm^2^) and the EPNs watered in with 560 mL (6.3 mm water) using a watering can. Untreated control plots received water only. These plots did not receive infested blueberries. EPN persistence was measured by baiting soil samples from each plot with wax moth larvae. Soil samples were taken just after EPN application, and at 7 and 21 days after application. On each sampling day, four soil cores (2.5 cm diameter × 5 cm depth) were taken with an Oakfield sampler per arena, combined and sealed in a plastic bag and brought to the laboratory. The soil cores were broken up and thoroughly mixed and a 25-g subsample combined with 75 g of pasteurized blueberry soil and placed in a 200-mL cup. Due to the at least initially high numbers of IJs expected per sample we only used 25 g per cup. As baiting of such a small soil amount with 10 wax moth larvae seemed difficult, we combined the sample soil with the additional pasteurized blueberry soil. After adding 10 wax moth larvae per cup, the samples were baited for four consecutive 3-day baiting rounds as described in the laboratory persistence experiment. After each 3-day period, any dead wax moth larvae were collected and replaced with fresh living ones. The dead larvae were rinsed in tap water and placed on moist filter paper for another 3 days before being dissected under a dissecting microscope to determine presence of EPNs. The cadavers were assigned to EPN species based on the cadaver color and morphology of any adult males within them [26,27]. Data were the total number of EPN-infected cadavers of the treatment EPN species recovered over the 12-day baiting period.

### 2.3. Field Experiment: 2021

Based on our findings in the 2020 field and laboratory experiments, we selected *S. riobrave* as the only EPN species for the 2021 field experiment. The experiment was conducted in two organically maintained blueberry fields at the Philip E. Marucci Center, Chatsworth, NJ, USA. In both fields, highbush blueberries were spaced 0.61 m between bush crowns and 5.49 m between treated rows center-to-center. Field 1 contained 7-year-old bushes of mix varieties, whereas field 2 contained 25-year-old bushes of the variety ‘Duke’. The methodology used for both efficacy and persistence determination was the same as used in the 2020 experiment unless otherwise mentioned. In 2021, *C. nenuphar* larvae rather than infested berries as in 2020 were used to infest the plots in hopes of increasing the number of adults emerging and reducing variability in adult emergence. One day prior to treatment applications, 50 *C. nenuphar* larvae (fourth instar) were added to the soil surface of each efficacy plot and allowed to dig into the soil. Treatments consisted of 0, 25, and 50 *S. riobrave* IJs/cm^2^ with seven replicates per treatment in each field for both efficacy and persistence determination. Treatments were applied on 8 June 2021 (10:00–12:00 EST, sunny, 25–30 °C) in field 1 and on 16 June 2021 (10:00–12:00 EST, sunny, 25–30 °C) in field 2. Field 2 was treated later because the fruit in this field ripened later than those in field 1. Weather data were obtained as described for the 2020 field experiment.

### 2.4. Statistical Analysis

Laboratory EPN persistence data were subjected to analysis of variance (ANOVA) and Tukey test for means separation. Data were analyzed by soil type with experiment repetition, EPN species, and baiting date as factors. For the laboratory virulence experiments, a generalized linear model (glm) with a binomial distribution corrected by Bonferroni–Holm test was used to analyze the effects of rates, EPN species (EPNs), and EPN species × rates on the *C. nenuphar* mortality for both larva and pupa at 7 and 14 DAT.

For the 2020 field experiment, data were normalized by log (x + 1) transformation and subjected to ANOVA and LSD for means separation. For the 2021 field experiment, data were not normally distributed and could not be normalized through transformations. They were therefore first analyzed by field using Kruskal–Wallis non-parametric test followed by Dunn’s test for means separation. Since data in both fields followed the same pattern, we then analyzed the data combined for both fields. Field persistence data were normalized by square root transformations and subjected to repeated measures ANOVA and means separated with Tukey test. Differences among means in all experiments were considered significant at *p* < 0.05 (Appendix A).

## 3. Results

### 3.1. Nematode Persistence in Different Soil Types

Significant interactions between EPN species and days after treatment were detected in sandy loam soil (*df* = 12, 180; F = 2.54; *p* < 0.001) and acidic blueberry soil (*df* = 12, 180; F = 2.82; *p* < 0.001). In both soils, the number of infected waxworms (=infection) was always the highest in *S. riobrave* and the lowest in *H. bacteriophora*. However, *S. riobrave* infection was significantly higher than *S. carpocapsae* infection only on day 14 in sandy loam and day 28 in blueberry soil. *Steinernema carpocapsae* infection was always higher than *H. bacteriophora* infection in both soils and also higher than *S. feltiae* infection on days 0–28 in sandy loam and days 0, 7, and 28 in blueberry soil. *Steinernema feltiae* infection was significantly higher than *H. bacteriophora* infection on days 14–56 in sandy loam and days 7–56 in blueberry soil.

In the sandy loam (Figure 2), all EPN species declined significantly over time. However, the decline occurred the quickest in *H. bacteriophora* with a significant decrease compared to 0 DAT already at 7 DAT and the lowest infection already reached at 28 DAT. The three *Steinernema* spp. followed the same pattern with the first significant decrease observed at 28 DAT and the lowest infection at 56 DAT.

In the blueberry soil (Figure 2), infection declined the quickest in *H. bacteriophora*, following the same pattern as in sandy loam, with the first significant decrease at 7 DAT and the lowest infection already at 28 DAT. In the *Steinernema* spp., decline tended to be slower than in sandy loam, but a significant decrease was observed in all three species at 28 DAT. However, only *S. riobrave* further declined by 56 DAT.

### 3.2. Virulence of EPNs to C. nenuphar Larvae

No control mortality was observed at 7 and 14 DAT, and the untreated control was not included in the analysis. At 7 DAT, mortality was significantly affected by EPN species (*df* = 3, 162; F = 49.35; *p* < 0.001) and rate (*df* = 2, 162; F = 15.95; *p* < 0.001); the factors did not interact (*df* = 6, 156; F = 0.70; *p* = 0.65). Mortality increased with rate except that no mortality was observed with *S. scarabaei* (Table 1). *Steinernema riobrave* caused the highest mortality followed by *S. carpocapsae*, whereas *S. feltiae* caused only minimal mortality. At 14 DAT, mortality was significantly affected by EPN species (*df* = 3, 162; F = 72.78; *p* < 0.001) and rate (*df* = 2, 162; F = 12.98; *p* < 0.001); the factors did not interact (*df* = 6, 156; F = 1.03; *p* = 0.41). At 14 DAT, mortality also increased with rate and *S. riobrave* caused the highest mortality followed by *S. carpocapsae* and *S. feltiae*, whereas *S. scarabaei* caused the lowest mortality.

### 3.3. Virulence of EPNs to C. nenuphar Pupae

In the first experiment comparing different EPN species, no control mortality was observed at 7 and 14 DAT, and the untreated control was not included in the analysis. At 7 DAT, ANOVA indicated significant differences among treatments (*df* = 3, 140; F = 3.62; *p* < 0.05). Only *S. feltiae* caused mortality, but also only at 4%. At 14 DAT, mortality was significantly affected by species (*df* = 3, 140; F = 9.64; *p* < 0.001) (Figure 3). Mortality was higher in the *S. carpocapsae* (50%) treatment than all others, *H. bacteriophora* (0%), *S. scarabaei* (17%), and *S. feltiae* (13%) (Figure 3).

In the second experiment comparing different *S. riobrave* rates, no control mortality was observed at 7 and 14 DAT, and the untreated control was not included in the analysis. At 7 DAT, mortality was already high at all rates (84–98%) but was significantly higher at the medium and high rate than at the low rate (*df* = 2, 237; F = 6.20; *p* < 0.001). At 14 DAT, very few additional pupae had been killed but total mortality was significantly higher at the medium and high rates than at the low rate (*df* = 2, 237; F = 12.07; *p* < 0.001) (Figure 4).

### 3.4. Field Experiments: 2020 and 2021

During the 2020 experiment comparing different EPN species (23 June–10 August 2020), the average (in parentheses absolute) daily high and low temperatures were 31.3 (35.6) °C and 19.3 (15.0) °C, respectively. Rainfall totaled 240 mm and was supplemented with 53 mm overhead irrigation. *Steinernema riobrave* provided significantly higher levels of *C. nenuphar* suppression than *S. carpocapsae*, *S. feltiae*, and *S. scarabaei* (*df* = 4, 29; F = 3.80; *p* < 0.05). There was no difference in *C. nenuphar* adult emergence between *S. carpocapsae*, *S. feltiae*, and *S. scarabaei* treatments and the untreated control (Figure 5).

During the 2021 experiment comparing *S. riobrave* rates, the average (in parentheses the absolutes) daily high and low temperatures for the experimental period in field 1 (8 June–15 July 2021) were 29.1 (35.0) °C and 17.5 (8.3) °C, respectively. Rainfall totaled 202 mm and was supplemented with 39 mm overhead irrigation. In field 2 (16 June–15 July 2021), average daily high and low temperatures were 29.7 (35.0) °C and 17.6 (8.3) °C, respectively. Rainfall totaled 149 mm and was supplement with 33 mm of overhead irrigation. Significantly more *C. nenuphar* adults emerged in the untreated control than at both *S. riobrave* rates (*df* = 2, 41; F = 17.51; *p* < 0.001). The low and high rates provided 80% and 100% control, respectively (Figure 5).

### 3.5. Nematode Field Persistence: 2020 and 2021

In 2020, no waxworms were infected in soil samples from the untreated control plots, which were hence not included in the analysis. EPN species (*df* = 6, 71; F = 17.42; *p* < 0.01) and sampling day (*df* = 2, 71; F = 140.31; *p* < 0.001) significantly affected the number of infected waxworms in the soil samples, but the two factors interacted significantly (*df* = 6, 71; F = 3.80; *p* < 0.01). Infection declined significantly over time in all EPN species (Figure 6). However, *S. feltiae* showed the greatest decline between 0 and 7 DAT with no more significant decline thereafter, whereas the other species declined less dramatically between 0 and 7 DAT but continued to decline until 21 DAT. Infections were the highest for *S. riobrave*, higher than for *S. feltiae* and *S. scarabaei* at 0 to 21 DAT, but higher than *S. carpocapsae* only at 21 DAT. *Steinernema feltiae* had the lowest number of infections at 7 DAT but did not differ from *S. scarabaei* and *S. carpocapsae* at 21 DAT.

In 2021, 2.3 ± 0.2, 1.4 ± 0.1, and 2.3 ± 0.2 waxworms were infected in samples from the untreated control plots at 0, 8, and 21 DAT, respectively. All these infections were caused by the native *Steinernema glaseri*, and the untreated control was therefore not included in the analysis. In the treated plots, all infections were caused by *S. riobrave*, and *S. riobrave* rate (*df* = 1, 83; F = 34.51; *p* < 0.001), sampling day (*df* = 2, 83; F = 53.85; *p* < 0.001), and experimental field (*df* = 1, 83; F = 6.91; *p* < 0.05) significantly affected the number of infected waxworms. Rate did not interact with field (*df* = 1, 83; F = 2.31; *p* = 0.11) but with sampling day (*df* = 2, 83; F = 3.82; *p* < 0.05). More *S. riobrave* infections were found on each sampling day at the high rate than at the low rate (Figure 6). However, infections declined progressively at the high rate but only declined from 0 to 8 DAT at the low rate and not thereafter.

## 4. Discussion

Our study clearly demonstrates the superiority and great potential of the EPN *S. riobrave* for the management of *C. nenuphar* in highbush blueberries. Among five EPN species tested it was the most persistent in blueberry soil, was the most virulent against fourth-instar *C. nenuphar* larvae entering the soil and pupae in laboratory studies and provided the highest level of control in field studies, with excellent control even at relatively low rates.

Under laboratory conditions, persistence of *S. riobrave* was high in sandy loam and acidic sand soils and its recovery did not start to decline until 28 DAT in both soils. *Steinernema carpocapsae* and *S. feltiae* demonstrated an intermediary persistence and their recovery started to decline slightly after 14 DAT in both soils. Persistence of *H. bacteriophora* was much lower in both soil types. Generally, *Steinernema* spp. appear to persist longer than *Heterorhabditis* spp. under laboratory and field conditions [28]. According to Koppenhöfer and Fuzy [20], the efficacy of *H. bacteriophora* appears to be generally lower in acidic soil than in common agricultural and turfgrass soils with a slightly acidic to neutral pH. In our study, we observed that the persistence and efficacy of this nematode was lower not only in acidic blueberry soil but also in sandy loam soil with a pH of 6–6.5. Our 2020 field trial results are consistent with our laboratory results. *Steinernema riobrave* also demonstrated the highest persistence (=infections) than the others *Steinernema* spp. In 2021, at the high rate, *S. riobrave* infections declined on each sampling day, and at the low rate the infections declined from 0 to 8 DAT but not thereafter. Even with the decline for both rates, *S. riobrave* demonstrated excellent persistence. Although there are many factors that can affect EPN performance in different soil types—such as weather conditions, soil particle size composition, pH, organic matter, soil nutrient concentration, length of exposure, water potential, etc. [20,29]—in our study, our focus was to determine the best EPN species to persist and control *C. nenuphar* in highbush blueberry.

Our laboratory and field results are consistent about *S. riobrave* performance, and they are similar to previous findings that also indicated *S. riobrave* to be highly effective in suppressing *C. nenuphar* in soil applications. In the laboratory, we observed superior virulence of *S. riobrave* against *C. nenuphar* larvae and pupae, and in the field, *S. riobrave* provided significantly higher levels of *C. nenuphar* suppression (90%) than *S. carpocapsae*, *S. feltiae*, and *S. scarabaei*. Studies conducted with *S. riobrave* in apple and cherry orchards, in peach orchards and in wild plum thickets located in Michigan, Georgia, and Florida, reported 80–89%, 78–100%, and >94% *C. nenuphar* control, respectively [11,12,17]. Additionally, Shapiro-Ilan et al. [14] observed 85% and 97% *C. nenuphar* control with *S. riobrave* in apple orchards in Massachusetts and 100% control in West Virginia in two years of experiments. In our 2021 study, we confirmed the efficacy of *S. riobrave*, at low and high rates (25 and 50 IJs/cm^2^), against *C. nenuphar* in blueberry fields and no difference was detected in *C. nenuphar* suppression between the rates. Thus, our research extends that of previous findings in a variety of cropping systems and geographic locations and, also indicates that *S. riobrave* is highly effective in suppressing larvae and pupae of *C. nenuphar* in highbush blueberries with acidic blueberry sand in New Jersey. To date, *S. riobrave* is the most effective EPN tested for *C. nenuphar* control.

Regarding the other EPN species tested, *S. carpocapsae* and *S. feltiae* caused up to around 62% mortality of *C. nenuphar* larvae in the laboratory, albeit only at high rates, whereas *S. scarabaei* caused only limited mortality. Against pupae, *S. carpocapsae* was the only other species that caused significant mortality albeit much lower and slower than *S. riobrave*. Our laboratory results corroborate previous laboratory research that indicated that *C. nenuphar* is susceptible to various EPN species, such as *S. carpocapsae* and *S. feltiae* [8,10,11,30]. On the other hand, our field observations in blueberry fields differed from previous studies in other commodities because *S. carpocapsae* and *S. feltiae* did not suppress *C. nenuphar* at all, whereas *S. riobrave* was highly effective. Other studies have reported variable and reduced impacts of *S. carpocapsae* and *S. feltiae* on *C. nenuphar* in different crops. For example, Pereault et al. [17] observed that *S. carpocapsae* was less effective than *S. riobrave* in apple and cherry orchards in Michigan. Alston et al. [30] observed that *S. feltiae* applications produced low (22–39%) levels of *C. nenuphar* control in wild plum sites in Utah, and Shapiro-Ilan et al. [11] observed no suppression when using *S. feltiae* in Georgia and Florida peach orchards. Additionally, Shapiro-Ilan et al. [14] observed an intermediate level of control by *S. feltiae*, but the level of suppression was not consistent and was lower than the level of control provided by *S. riobrave* in apple orchards located in West Virginia and Massachusetts. The greater lag of *S. carpocapsae* and *S. feltiae* in our studies may be related to the better ability of *S. riobrave* to persist and to infect hosts in the highly acidic blueberry sand that adds to its higher virulence.

*Steinernema scarabaei* was included in our study because it can provide excellent control (86–95%) of another important pest of highbush blueberries, the larvae of *A. orientalis* [24]. In addition, laboratory data suggest that the low pH of typical blueberry soil has no negative effect on *S. scarabaei* persistence [20] and this EPN may therefore have good potential for long-term white grub suppression in blueberries [24]. Thus, a *S. scarabaei* application in June for *C. nenuphar* control may also control *A. orientalis* larvae appearing in the soil in July. However, in our study, this EPN species demonstrated limited potential as a *C. nenuphar* control agent in both laboratory and field experiments.

## 5. Conclusions

Our laboratory results match with our field findings and show that *S. riobrave* is, by far, the most effective EPN species to suppress *C. nenuphar* larvae and pupae in acidic blueberry soils. According to Shapiro-Ilan et al. [31], the use of *S. riobrave* as a component of an integrated program that targets multiple stages of *C. nenuphar* may be feasible. We here showed that *S. riobrave* can infect both larvae and pupae of *C. nenuphar* at different rates and is also very persistent in acidic blueberry soil, indicating that applications of this nematode species could control multiple stages (larvae and pupae) of *C. nenuphar* in the typically highly acidic blueberry sands. Future research should examine optimal timing of EPN application and combination with other management tools to develop and implement a multi-stage integrated pest management program for *C. nenuphar* in highbush blueberries.

## Figures and Tables

**Figure 1 biology-11-00045-f001:**
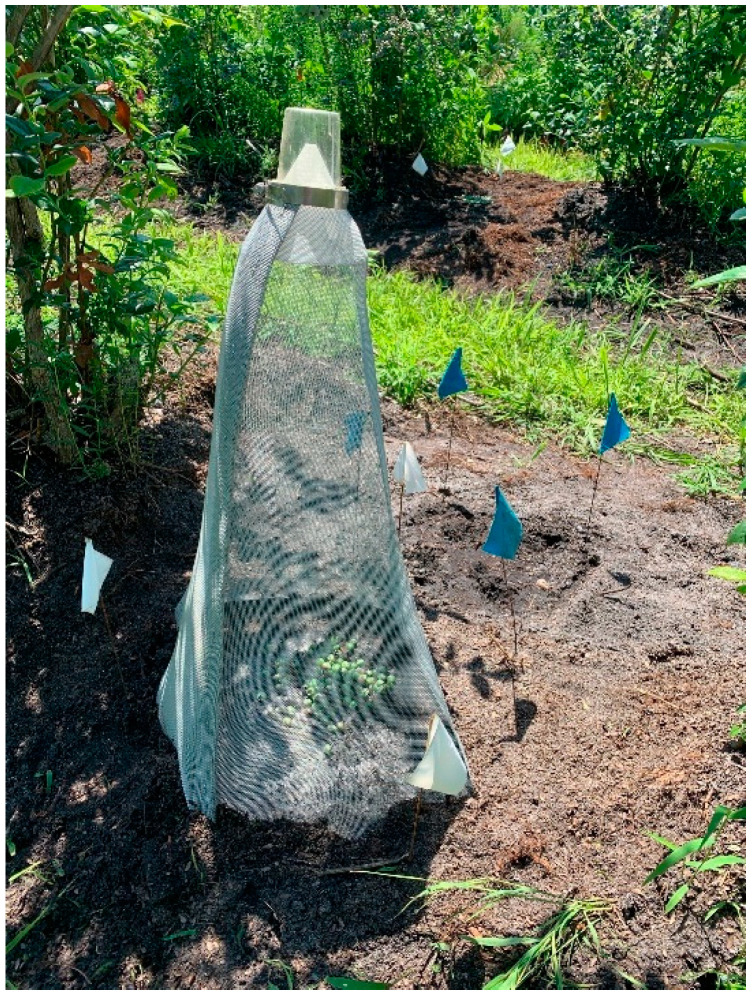
Pyramidal emergence cage (43.7 cm diameter base; 64 cm tall) made of aluminum window screen and a boll weevil trap fitted with a stainless-steel hose clamp at the top (lower left) and arenas (30 × 30 cm) for the determination of entomopathogenic nematode persistence (upper right).

**Figure 2 biology-11-00045-f002:**
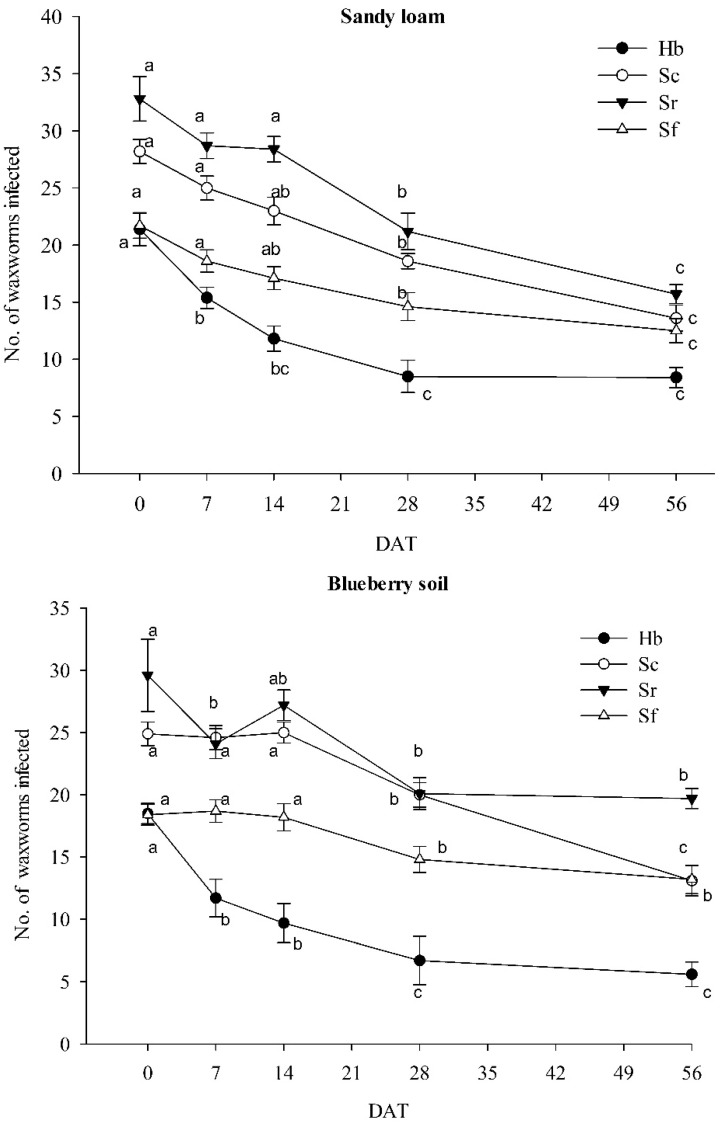
Persistence of the entomopathogenic nematodes *Heterorhabditis bacteriophora* (Hb), *Steinernema carpocapsae* (Sc), *S. riobrave* (Sr), and *S. feltiae* (Sf) in cups with 100 g of moist sandy loam (**top**) and blueberry soil (**bottom**). Persistence was determined by saturation baiting with waxworms, the larvae of the greater wax moth. Letters next to symbols indicate significant differences in the total number of waxworms infected among baiting dates within species (*p* < 0.05).

**Figure 3 biology-11-00045-f003:**
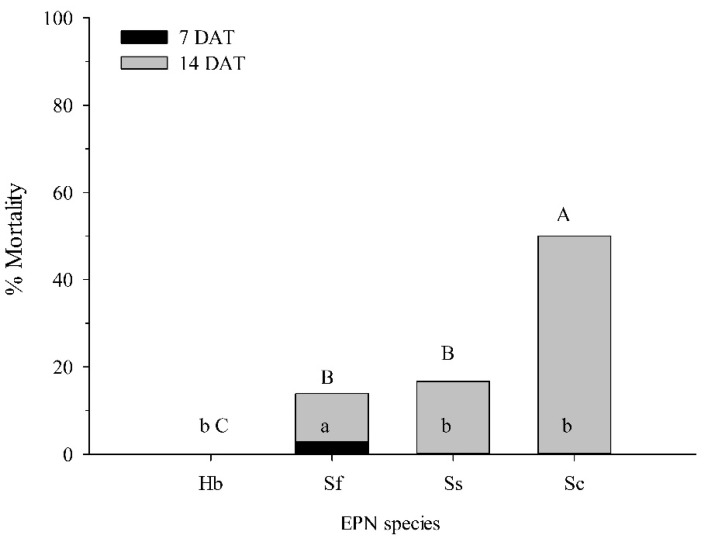
Percentage mortality (±SEM) of *Conotrachelus nenuphar* pupae at 7 and 14 days after treatment (DAT) with 400 infective juveniles of the entomopathogenic nematodes *Steinernema carpocapsae* (Sc), *S. feltiae* (Sf), *S. scarabaei* (Ss), and *Heterorhabditis bacteriophora* (Hb) in 30-mL cups with 20 g moist blueberry soil and one pupa. Bars with the same lower and uppercase letter do not differ significantly at 7 and 14 DAT, respectively (*p* < 0.05).

**Figure 4 biology-11-00045-f004:**
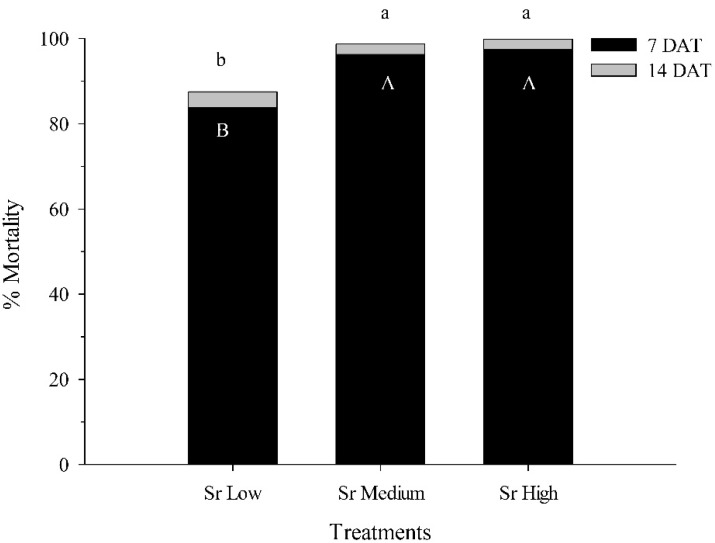
Percentage mortality (±SEM) of *Conotrachelus nenuphar* pupae at 7 and 14 days after treatment (DAT) with three rates (100, 200, 400 infective juveniles) of the entomopathogenic nematodes *Steinernema riobrave* (Sr) in 30-mL cups with 20 g moist blueberry soil and one pupa. Bars with same uppercase and lowercase letters do not differ significantly at 7 and 14 DAT, respectively (*p* < 0.05).

**Figure 5 biology-11-00045-f005:**
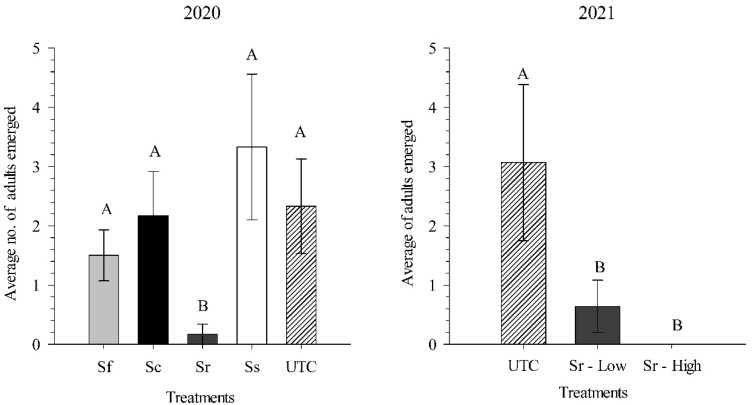
Average number (±SEM) of adult *Conotrachelus nenuphar* emerged from field plots following applications of the entomopathogenic nematodes *Steinernema feltiae* (Sf), *S. carpocapsae* (Sc), *S. riobrave* (Sr), and *S. scarabaei* (Ss) (all at 50 infective juveniles per cm^2^) in 2020 and following applications of a high and low rate (25 and 50 infective juveniles per cm^2^) of *S. riobrave* in 2021. UTC = untreated control. Bars with the same letters are not statistically different (*p* < 0.05).

**Figure 6 biology-11-00045-f006:**
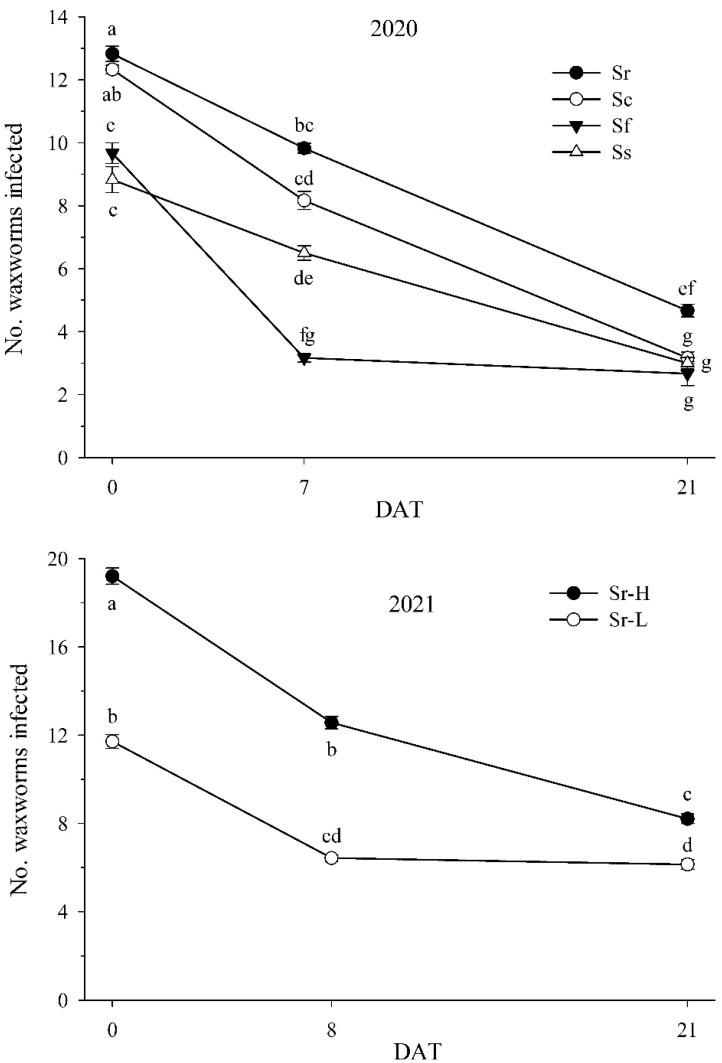
Persistence of the entomopathogenic nematodes *Steinernema carpocapsae* (Sc), *S. riobrave* (Sr), *S. feltiae* (Sf), and *S. scarabaei* (applied at 50 infective juveniles per cm^2^) in the 2020 field experiment (**top**) and of *S. riobrave* applied at a low (-L; 25 per cm^2^) and high (-H; 50 per cm^2^) rate in the 2021 field experiment (**bottom**). Persistence was determined by saturation baiting with waxworms of 100 g soil samples taken from the field plots 0 to 21 days after treatment (DAT). Letters next to symbols indicate significant differences in the total number of waxworms (*p* < 0.05).

**Table 1 biology-11-00045-t001:** Percentage mortality (±SEM) of *Conotrachelus nenuphar* larvae exposed to three rates ^#^ of four entomopathogenic nematode species at 7 and 14 days after treatment (DAT).

IJs/larva	*S. feltiae*	*S. carpocapsae*	*S. riobrave*	*S. scarabaei*
7 DAT				
100	1.4 ± 1.4 Cb	2.4 ± 2.4 Bc	30 ± 8.1 Ac	0.0 ± 0.0 Da
200	1.4 ± 1.4 Cb	8.6 ± 4.6 Bb	55.7 ± 9.4 Ab	0.0 ± 0.0 Da
400	2.9 ± 2.9 Ca	35.7 ± 10.5 Ba	78.6 ± 7.4 Aa	0.0 ± 0.0 Da
14 DAT				
100	34.3 ± 8.0 Bc	41.4 ± 5.7 Bc	94.3 ± 2.3 Ab	7.1 ± 2.7 Cc
200	41.4 ± 6.4 Bb	55.7 ± 8.1 Bb	95.7 ± 2.5 Aa	27.1 ± 6.2 Cb
400	62.9 ± 7.2 Ba	61.4 ± 8.8 Ba	98.6 ± 1.4 Aa	37.1 ± 4.6 Ca

Uppercase letters indicate significant differences between species within rate and DAT (*p* < 0.05). Lowercase letters indicate significant differences between rates within species and DAT (*p* < 0.05). ^#^ Five larvae per cup with 100 g moist blueberry soil exposed to 500, 1000, or 2000 infective juvenile nematodes per cup.

## Data Availability

The data presented are available in the Appendix A.

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
