# Peer review of "Entomopathogenic Nematodes for the Management of Plum Curculio in Highbush Blueberry"

_biology, 2021, doi:10.3390/biology11010045_

Round 1

Reviewer 1 Report

Manuscript ID: 1505369

Entomopathogenic Nematodes for the Management of Plum Curculio in Highbush Blueberry

               Interesting research topic. In times of intensive reduction of plant protection products, the natural defense mechanisms of plants are the solution to the problem. Entomopathogenic nematodes for the management of insects are known for many years. However, in this manuscript new possibilities were proposed.

               This is very well organized manuscript. I found this “ms” interesting and innovative. However, a few (ONLY) questions must be explained more precisely.

Critical review:

  1. Introduction in present form can’t be accepted. It looks very poor. I recommend to add more RECENT scientific literature. A lot of cited literature is rather old.
  2. What about the release of volatile organic compounds by plants or the production of phytoecdysteroids in context of plant protection in the future. No information was presented. It would be of importance at least mention about different ways of non-chemical plant protection.
  3. Methods, Results and Discussion are very clear and well presented.
  4. Figures are very legible.

Some other papers to add:

Pulsed Odors from Maize or Spinach Elicit Orientation in European Corn Borer Neonate Larvae

Journal of Chemical Ecology 35, 1032–1042 (2009)

DOI: 10.1007/S10886-009-9676-7

Orientation of European corn borer first instar larvae to synthetic green leaf volatiles

Journal of Applied Entomology 137(3), 234-240 (2013)

DOI: 10.1111/J.1439-0418.2012.01719.X

Author Response

Reviewer #1 (R1):

R1:  Introduction in present form can’t be accepted. It looks very poor. I recommend to add more RECENT scientific literature. A lot of cited literature is rather old.

   AU:  The reviewer makes a strong statement about the intro (which the other 2 reviewers have no problems with) but does not elaborate beyond the literature supposedly being outdated.  We feel that the intro in general is quite appropriate for the article.  Hence, I can only assume that the reviewer is referring to several older references used in the first paragraph because the literature cited in the remaining intro is not that old and quite up-to-date.  Of the 8 articles cited in the first paragraph, the years of publication are 1992, 2005, 2009, 1992, 1977, 1912, 2020, and 1993.  Indeed there are several articles from the 1990s and two even older ones.  But that comes with the nature of this section: the biology of the pest.  Since the biology was described very well and for the first time in these older papers it is very appropriate to cite these papers.  If the reviewer is aware of newer papers that describe the pest biology better, we would be happy to cite them.

R1: What about the release of volatile organic compounds by plants or the production of phytoecdysteroids in context of plant protection in the future. No information was presented. It would be of importance at least mention about different ways of non-chemical plant protection.

   AU:  If we were to mention these, we should then also mention a whole bunch of other management approaches and then also give references for them as the reviewer is suggesting for the VOCs.  But this is not a review paper, and we believe that mentioning all these other approaches here is outside of the scope of the paper and not necessary to better introduce the study.

Reviewer 2 Report

I have now completed my review of research ms No. biology-1505369, this ms reports very interesting and the authors intend to evaluate the efficacy of different entomopathogenic nematode (EPN) species both in lab and two-year field trails against plum snout beetle, Conotrachelus nenuphar. Steinernema riobrave exhibited highest virulence against larvae and pupae in lab – the same species of EPN effectively suppressed the C. nenuphar in field trials. The authors generated good labd and field data and claimed that to date, no EPNs have been tested for the control of this insect pest in highbush blueberries.

The objectives of the study are clear, the experimental design is appropriate and the results support the conclusion. Overall, the article is well written and based on the merits of study I am convinced this ms can be accepted for its publication in Biology with following few minor changes:

L23-37: Abstract should be different and more technical from Simple Summary

L131: how you avoided webbing in larvae of wax moths?

L153-160: why not all the EPN species were tested in single experiment and why different doses were used for S. riobrave?

L184: replace “by (18)” with “Piñero et al. (18)”

L236-238: In year 2020 the infested berries were used but next year directly infested larvae were used in plots, why the infestation method had been changed?

L250: give reference for “Tukey test”

L256: why authors did not normalized the year 2021 field data using log (x + 1), same procedure should be followed for analyzing the data of both years?

During review I found following most releveant and recent literature, the authors may consider to cite them in the ms

Gulzar, S., Wakil, W., Shapiro-Ilan, D.I. 2021. Potential use of entomopathogenic nematodes against the soil dwelling stages of onion thrips, Thrips tabaci Lindeman: Laboratory, greenhouse and field trials. Biological Control. 161: 104677. doi.org/10.1016/j.biocontrol.2021.104677

Usman, M., W. Wakil, D.I. Shapiro-Ilan. 2021. Entomopathogenic nematodes as biological control agent against Bactrocera zonata and Bactrocera dorsalis (Diptera: Tephritidae). Biological Control. 163: 104706. doi.org/10.1016/j.biocontrol.2021.104706

Author Response

Reviewer #2 (R2):

L23-37: Abstract should be different and more technical from Simple Summary.

   AU:  Why does the Abstract have to be so different from the Simple Summary, and why (and how) should it be more technical than it is?  It represents well was done in the study.  We could give some more details of the lab studies like percentage mortalities if we were allowed more words.  But we are already at the word limit.  And since neither of the 3 reviewers sees any issues with the Simple Summary, there seems to be no point in changing that to be more different than the Abstract.

L131: how you avoided webbing in larvae of wax moths?

   AU:  By getting them from a commercial source!  I assume they heat treat the larvae which I remember reading about as the method to avoid the webbing.  I do not remember the details for that as I never had to do that.  I have only reared wax moths on my own to obtain pupae and adults for which the webbing was no issue.

L153-160: why not all the EPN species were tested in single experiment and why different doses were used for S. riobrave?

   AU: We did not have enough pupae at the time to include Sr in the first experiment.  And based on the finding of the experiments with larvae we assumed that Sr would be more virulent to pupae than the other EPN species and hence included two lower dosages.  If I remember correctly, we also did a small preliminary experiment that indicated the higher virulence of Sr.

L184: replace “by (18)” with “Piñero et al. (18)”

   AU:  done.

L236-238: In year 2020 the infested berries were used but next year directly infested larvae were used in plots, why the infestation method had been changed?

   AU:  We were not happy with the low number of adults emerging in the first field experiment using infested berries as inoculum.  Based on suggestions from colleagues that did similar experiments with plum curculio in other systems, we therefore tried the release of already emerged larvae.  With half as many larvae in 2021 as berries in 2020 we got about the same number of adults in the UTC and the variability was not any different.  And releasing larvae rather than infested berries is of course more work.  Next time we would go back to berries as that seems less work and more natural.  Anyway, we added a sentence to explain the change.

L250: give reference for “Tukey test”

   AU:  Tukey test is such a common procedure.  Why do we need a reference for that?  The other reviewers had not issue with this.

L256: why authors did not normalized the year 2021 field data using log (x + 1), same procedure should be followed for analyzing the data of both years?

   AU:  We added a few words to explain that 2021 data could not be normalized through transformations incl. the one suggested and had hence be analyzed as described.

R2:  During review I found following most relevant and recent literature, the authors may consider to cite them in the ms:

Gulzar, S., Wakil, W., Shapiro-Ilan, D.I. 2021. Potential use of entomopathogenic nematodes against the soil dwelling stages of onion thrips, Thrips tabaci Lindeman: Laboratory, greenhouse and field trials. Biological Control. 161: 104677. doi.org/10.1016/j.biocontrol.2021.104677

Usman, M., W. Wakil, D.I. Shapiro-Ilan. 2021. Entomopathogenic nematodes as biological control agent against Bactrocera zonata and Bactrocera dorsalis (Diptera: Tephritidae). Biological Control. 163: 104706. doi.org/10.1016/j.biocontrol.2021.104706

   AU:  Not sure how these articles are relevant.  They use EPNs for pest control, but against very different insect types in very different systems.  We prefer not to add that as people would then just wonder why we are adding these and not a whole bunch of others.

Reviewer 3 Report

Some minor changes are proposed. 

See comments in the ms.

Author Response

Reviewer #3 (R3)All comments in ms file.

L2-3:  R3 suggests changing “Management” to “Control”.

   Au: Why not management?  We prefer management.

L21 & 36:  R3 suggests “integrated pest management” instead of “management”. 

   Au: We believe that this approach could also be a part of management that does not necessarily fall under IPM. Hence, we prefer to keep just “management”.

L26.  R3 wonders whether the years is correct.

   Au:  2020 is correct here.  The 2021 part comes in L35. 

L38:  R3 suggest to add “nematodes” as a key word.

   Au:  “nematodes” is extremely broad, like “insects” would be.  We could add “entomopathogenic nematodes” but that is already in the title so not necessary.